# Determination of Serum Progranulin in Patients with Untreated Familial Hypercholesterolemia

**DOI:** 10.3390/biomedicines10040771

**Published:** 2022-03-25

**Authors:** Bíborka Nádró, Hajnalka Lőrincz, Lilla Juhász, Anita Szentpéteri, Ferenc Sztanek, Éva Varga, Dénes Páll, György Paragh, Mariann Harangi

**Affiliations:** 1Department of Internal Medicine, Division of Metabolic Disorders, Faculty of Medicine, University of Debrecen, 4032 Debrecen, Hungary; nadro.biborka@med.unideb.hu (B.N.); lorincz_hajnalka@belklinika.com (H.L.); lilla.juhasz@med.unideb.hu (L.J.); szentpeteri.anita@gmail.com (A.S.); sztanek@med.unideb.hu (F.S.); pall.denes@med.unideb.hu (D.P.); paragh.gyorgy@med.unideb.hu (G.P.); 2Doctoral School of Health Sciences, Faculty of Public Health, University of Debrecen, 4032 Debrecen, Hungary; 3Department of Internal Medicine and Hematology, Semmelweis University, 1088 Budapest, Hungary; varga.eva@med.semmelweis-univ.hu; 4Department of Medical Clinical Pharmacology, Faculty of Medicine, University of Debrecen, 4032 Debrecen, Hungary

**Keywords:** progranulin, familial hypercholesterolemia, inflammation, tumor necrosis factor alpha, high-density lipoprotein subfraction

## Abstract

Background: Familial hypercholesterolemia (FH) is an autosomal dominant trait characterized by elevated LDL-C concentrations and is associated with an increased risk of premature atherosclerosis. Progranulin (PGRN) is a multifunctional protein that is known to have various anti-atherogenic effects. To date, the use of serum PGRN in patients with FH has not been studied. Methods: In total, 81 untreated patients with heterozygous FH (HeFH) and 32 healthy control subjects were included in this study. Serum PGRN, sICAM-1, sVCAM-1, oxLDL and TNFα concentrations were determined by ELISA. Lipoprotein subfractions were detected by Lipoprint. We diagnosed FH using the Dutch Lipid Clinic Network criteria. Results: We could not find a significant difference between the PGRN concentrations of the HeFH patients and controls (37.66 ± 9.75 vs. 38.43 ± 7.74 ng/mL, ns.). We found significant positive correlations between triglyceride, TNFα, sVCAM-1, the ratio of small HDL subfraction and PGRN, while significant negative correlations were found between the ratio of large HDL subfraction and PGRN both in the whole study population and in FH patients. PGRN was predicted by sVCAM-1, logTNFα and the ratio of small HDL subfraction. Conclusions: The strong correlations between HDL subfractions, inflammatory markers and PGRN suggest that PGRN may exert its anti-atherogenic effect in HeFH through the alteration of HDL composition and the amelioration of inflammation rather than through decreasing oxidative stress.

## 1. Introduction

Familial hypercholesterolemia (FH) is a genetic disorder affecting lipoprotein metabolism and is a common inherited autosomal co-dominant disorder characterized by high plasma cholesterol and low-density lipoprotein cholesterol (LDL-C). Formerly, the prevalence of FH in its heterozygous form (HeFH) has been considered to be approximately 1:500. However, recent meta-analyses have revealed that its prevalence rates are around 1:300 in the general population. Homozygous FH (HoFH) has an estimated prevalence of one:160,000 to 500,000 [1]. Based on our previous study using data mining and calculated Dutch Lipid Clinic Network (DLCN) scores, the prevalence of HeFH in our country is also about 1:340 [2]. The markedly increased levels of serum total cholesterol and LDL-C in FH patients promote early atherosclerosis, leading to premature cardiovascular disease and an increased risk of myocardial infarction, stroke and peripheral vascular disease.

Atherosclerosis is a complex multifactorial disease with a chronic and progressive pathogenesis, which is characterized by lipid accumulation and low-grade inflammation. Athero-inflammation involves the activation of both innate and adaptive immune responses, leading to the progression and destabilization of atherosclerotic plaque [3].

As well as lipid parameters and the other traditional risk factors of atherosclerosis, the pathogenic role of some further oxidative and inflammatory proteins has also emerged in the last few decades [4]. However, the regulation of their expression has not been fully elucidated. 

Progranulin (PGRN) is a growth factor that has been shown to be expressed in many tissues, including epithelia, myeloid- and lymphoid-derived cell lines, bone marrow and solid organs [5]. This cysteine-rich, glycosylated protein has initially been identified as a growth factor that is involved in numerous physiological and pathological processes, such as inflammation, neuronal cell growth, tumorigenesis and wound healing [6,7]. Previously, PGRN has also been detected in human and murine atherosclerotic plaque [8,9]. PGRN is expressed by multiple cell types of atherosclerotic lesions, including macrophages [10], smooth muscle cells [8,9] and endothelial cells [11]. A global deficiency of PGRN increases atherosclerosis in ApoE^−/−^ mice [9], although the mechanisms responsible for this phenotype are unclear. Previously, it has been hypothesized that immune cell-derived PGRN modulates the development of murine atherosclerosis [12]. To date, the serum concentrations of PGRN in patients with HeFH have not been studied. Therefore, we aimed to measure PGRN in the sera of patients with untreated HeFH. 

We hypothesized that serum PGRN is higher in untreated patients with HeFH and that its serum concentration correlates with inflammatory, oxidative and lipoprotein parameters.

## 2. Materials and Methods

### 2.1. Study Population

We enrolled 81 patients (females and males) with HeFH and 33 age- and gender-matched healthy control subjects. Patients were recruited from the Lipid Outpatient Clinic of the Department of Internal Medicine, University of Debrecen. All HeFH patients were either heterozygous with a confirmed LDL receptor gene mutation or fulfilled the Dutch Lipid Clinic Network diagnostic criteria for FH [13]. The patients were referred to our Lipid Outpatient Clinic by GPs and other specialists, such as cardiologists and neurologists, to verify the diagnosis of HeFH and initiate optimal therapy. These were scheduled medical appointments from 08:00–10:00 AM and we asked the patients to arrive after 12 h of fasting. All patients were newly diagnosed without ongoing lipid-lowering medical treatment. The sample of patients referred to our General Outpatient Clinic for routine wellbeing physicals was used as the control. Only patients that were taking no medications and had no previous chronic diseases or acute illnesses in the past 3 months were selected as controls. Furthermore, from the physical examination of the controls, their electrocardiograms and laboratory tests, including lipid parameters, were free of any abnormalities.

Patients with a previous history of type 1 or type 2 diabetes were excluded from the study, as well as subjects with alcoholism, known liver diseases, autoimmune or endocrine diseases and neurological or hematological disorders, which can be associated with peripheral polyneuropathy. Pregnant women or subjects with established malignancy were also excluded. All participants provided written informed consent. The study protocol was approved by the local and regional ethical committees (DE RKEB/IKEB 4775-2017, date obtained: 3 April 2020 and ETT TUKEB 34952-1/2017/EKU, date obtained: 30 June 2017) and the study was carried out in accordance with the Declaration of Helsinki.

### 2.2. Blood Sampling

Venous blood samples were drawn after overnight fasting and the sera were prepared immediately. Routine laboratory investigations (triglyceride, total cholesterol, low-density lipoprotein cholesterol (LDL-C), high-density lipoprotein cholesterol (HDL-C), lipoprotein(a) (Lp(a)), creatinine, uric acid and glucose) were carried out on the fresh sera using a Cobas c501 autoanalyzer (Roche Ltd., Mannheim, Germany) in the Department of Laboratory Medicine, Faculty of Medicine, University of Debrecen, Hungary. The sera to be used for enzyme activity measurements and ELISA determinations were kept at −70 °C before analysis. The reagents were purchased from the same vendor and the tests were performed according to the recommendations of the manufacturer.

### 2.3. Measurement of Progranulin and Asymmetric Dimethyl Arginine (ADMA)

Serum PGRN and ADMA concentrations were measured using the commercially available competitive enzyme-linked immunosorbent assay (ELISA) (BioVendor, Brno, Czech Republic and DLD Diagnostika GmBH, Hamburg, Germany), with intra-assay CVs ranging from 3.38 to 4.35% and inter-assay CVs ranging from 6.36 to 7.99% in the case of PGRN and intra-assay CVs of 4.9–5.4% and inter-assay CVs of 4.3–9.6% in the case of ADMA. The values were expressed as ng/mL and μmol/L, respectively.

### 2.4. Measurement of TNFα

Serum TNFα was assessed using the TNFα ELISA (R&D Systems Europe Ltd., Abington, UK). The measurement of serum TNFα was performed according to the recommendations of the manufacturer. The intra-assay CVs ranged from 1.9 to 2.2% and inter-assay CVs from 6.2 to 6.7%. The values were expressed as pg/mL.

### 2.5. Measurement of oxLDL

The serum concentrations of oxidized LDL (oxLDL) were detected busing the commercial sandwich ELISA (Mercodia AB, Uppsala, Sweden), which is based on a direct sandwich technique where two monoclonal antibodies are directed against separate antigenic determinants on the oxidized apolipoprotein B molecule. The sensitivity of the oxLDL measurements was < 1 mU/L and the intra- and inter-assay coefficients of variation were 5.5–7.3% and 4–6.2%, respectively. 

### 2.6. Measurement of sICAM-1, sVCAM-1 and sCD40L

The serum sICAM-1, sVCAM-1 and sCD40L were measured using the sandwich ELISA (R&D Systems Europe Ltd., Abington, UK). The ELISA procedures were carried out according to the manufacturer’s instructions. The intra-assay and inter-assay CVs ranged between 3.7–5.2% and 4.4–6.7% (ICAM-1), 2.3–3.6% and 5.5–7.8% (VCAM-1) and 4.5–5.4% and 6.0–6.4% (sCD40L). The values were expressed as ng/mL.

### 2.7. Measurement of Serum Myeloperoxidase

The serum concentrations of MPO were measured using the commercially available sandwich ELISA (R&D Systems Europe Ltd., Abington, UK). The intra- and inter-assay coefficients of variation were 6.5–9.4%. The ELISA assay was performed according to the manufacturer’s instructions.

### 2.8. Determination of PON1 Enzyme Activity

The serum PON1 arylesterase activity was measured using phenylacetate substrate (Sigma Aldrich, Budapest, Hungary) and the hydrolysis of the phenylacetate was monitored at 270 nm at room temperature, as formerly described in [14]. The serum PON1 paraoxonase activity was assayed on a microtiter plate via a kinetic semiautomated method using paraoxon (O,Odiethyl-O-p-nitrophenyl phosphate, Sigma-Aldrich, Budapest, Hungary) as a substrate. The hydrolysis of the paraoxon was followed at 405 nm at room temperature.

### 2.9. Lipoprotein Subfraction Analyses

The LDL and HDL lipoprotein subfractions were distributed using electrophoresis on polyacrylamide gel with the Lipoprint System (Quantimetrix Corporation, Redondo Beach, CA, USA), as previously described in [15]. Briefly, 25 µL of each patient’s serum sample was transferred to polyacrylamide gel tubes containing 200 and 300 µL of loading gel and lipophilic dye (Sudan Black, Sigma Aldrich, Budapest, Hungary). After 30 min of photopolymerization, the tubes were electrophorized using a 3 mA electric current. Each electrophoresis chamber involved a quality control provided by the manufacturer (Liposure Serum Lipoprotein Control, Quantimetrix Corporation, Redondo Beach, CA, USA). The subfraction bands were scanned with an ArtixScan M1 digital scanner (Microtek International Inc., Hsinchu, Taiwan) and analyzed with Lipoware software (Quantimetrix Corporation, Redondo Beach, CA, USA).

During the LDL subfraction analysis, up to seven LDL subfractions were distributed. The proportion of large LDL (large LDL %) was defined as the summed percentages of LDL1 and LDL2, whereas the proportion of small LDL (small dense LDL %) was defined as the sum of LDL3–LDL7. The cholesterol concentrations of LDL subfractions were determined by multiplying the relative area under the curve (AUC) of subfractions by the total cholesterol concentration. We calculated the total LDL-C as the sum of cholesterol in the IDL (MidA–C) and LDL subfractions (LDL1–LDL7), which correlated with the directly determined LDL–C. [16].

In the HDL subfraction tests, large (from HDL1 to HDL3), intermediate (from HDL4 to HDL7) and small (HDL8 to HDL10) subfractions were distributed between LDL and albumin peaks. The cholesterol contents of the HDL subfractions were calculated using the Lipoware software (Quantimetrix Corp., Redondo Beach, CA, USA), based on the relative AUC of the subfraction bands.

### 2.10. Statistical Methods

The statistical analyses were performed using the Statistica 13.5.0.17 software (TIBCO Software Inc. Palo Alto, CA, USA) and GraphPad Prism 6.01 (GraphPad Prism Software Inc., San Diego, CA, USA). We performed the statistical power analysis using the calculator of SPH Analytics (SPH Analytics LTD., Alpharetta, GA, USA) to validate the difference in serum PGRN concentrations between FH (group 1) and control individuals (group 2). The statistical power was above 0.8 (0.98). The relationship between the two categorical variables was calculated with a chi-squared test. The normality of distribution was tested by the Kolmogorov–Smirnov test. Comparisons between the groups were performed using Student’s unpaired *t*-test in the case of normally distributed variables and the Mann–Whitney U test in the case of variables with non-normal distribution. The data were expressed as mean ± SD or median (upper–lower quartiles). Pearson correlation was used to investigate the relationship between the selected variables. One-way ANOVA was performed to study the effects of VLDL and HDL-C on serum PGRN in HeFH patients and controls. A multiple regression analysis was performed to determine the best predicted PGRN concentrations of the variables. *p* ≤ 0.05 probability values were considered statistically significant.

## 3. Results

Significantly higher serum total cholesterol, LDL-C, triglyceride, apoB100 and Lp(a) were found in HeFH patients compared to the control subjects, while serum HDL-C and ApoA1 did not differ significantly. Although PON1 arylesterase activity was significantly higher in HeFH patients compared to the controls, we could not find any significant differences in PON1 paraoxonase or salt stimulated paraoxonase activities. Serum myeloperoxidase, oxLDL, sICAM-1 and TNFα were significantly higher in HeFH patients compared to the controls, while there were no significant differences in serum hsCRP, sVCAM-1, sCD40L or ADMA between the two study groups. Serum PGRN did not differ significantly between HeFH patients and the controls (Table 1).

The proportion and concentration of VLDL and the concentration of IDL subfractions were significantly higher in HeFH patients compared to the controls. Significantly higher small dense LDL ratios and concentrations, as well as lower mean LDL sizes, were also found in HeFH patients compared to the controls (Table 2). 

Lower ratios and concentrations of large and intermediate HDL subfractions, in contrast to the higher ratios and concentrations of small HDL subfractions, were found in HeFH patients compared to the control subjects (Table 2).

The HeFH patients were divided into high and low VLDL and high and low HDL-C subgroups according to median VLDL and HDL-C. Their serum PGRN was compared to that of the control population. Moderately, but not significantly, higher serum PGRN was found in HeFH patients with high VLDL compared to patients with low VLDL. Furthermore, we found a non-significant tendency for higher PGRN in HeFH patients with low HDL-C compared to patients with high HDL-C (Table 3).

We found significant positive correlations between age (r = 0.21; *p* = 0.03), the ratio of VLDL subfractions (r = 0.24; *p* = 0.02) and serum PRGN, while the correlation between PGRN and VLDL was on the border of statistical significance (r = 0.19; *p* = 0.057). Significant negative correlations were found between the ratio of IDL subfractions (r = −0.24; *p* = 0.01), mean LDL size (r = −0.32; *p* < 0.001) and PGRN in the whole study population (data not shown). Serum triglyceride correlated positively with PGRN both in HeFH patients and the whole study population (Figure 1a,c), but not in the controls (Figure 1b). A significant negative correlation was found between HDL-C and PRGN in the whole study population (Figure 1f), but not in the separate groups (Figure 1d,e). We found significant negative correlations between the ratios of large HDL subfractions and serum PGRN in all patient groups (Figure 2a–c), while there were significant positive correlations between the ratios of small HDL and serum PGRN in all populations (Figure 2d–f). Serum TNFα correlated positively with PGRN in HeFH patients, controls and in the whole study population (Figure 3a–c). There were significant positive correlations between serum sVCAM-1 and PGRN in HeFH patients and in the whole study population (Figure 3d,f), but not in the controls (Figure 3e).

We did not find significant correlations between serum oxLDL and PGRN in HeFH patients (r = 0.19; *p* = 0.11) or in the whole study population (r = 0.13; *p* = 0.20); however, there was a significant correlation in the control subjects (r = 0.45; *p* < 0.05).

Slightly, but not significantly, higher PGRN concentrations were found in patients with a positive history of acute myocardial infarction (AMI) (*n* = 5, mean age at first AMI: 53.8 ± 5.9 years) compared to patients without AMI (*n* = 75) (43.07 vs. 37.46 ng/mL, *p* = 0.06) (Figure 4). There were no differences in PGRN between patients with and without vascular complications, stroke and CAAD in their history (Figure 4).

A backward stepwise multiple regression analysis (the model contained age, logTG, HDL-C, ratio of large HDL, ratio of small HDL, logTNFα and sVCAM-1) showed that serum PGRN turned out to be best predicted by sVCAM-1 (β = 0.275; *p* < 0.001), logTNFα (β = 0.325; *p* < 0.001) and the ratio of small HDL subfractions (β = 0.301; *p* < 0.001).

## 4. Discussion

This is the first report to demonstrate the serum concentration of PGRN in patients with untreated familial hypercholesterolemia and its correlations with lipid parameters and the oxidative and inflammatory markers of atherosclerosis.

PGRN, also known as acrogranin, proepithelin, GP88, granulin–epithelin precursor (GEP) or PC cell-derived growth factor (PCDGF), is highly expressed in mammalian cells, including endothelial cells and macrophages, and it performs many biological functions [17]. PGRN has received attention as an important modulator of the inflammatory process as it binds directly to TNF receptors and disrupts TNFα signaling [18]. PGRN has been shown to inhibit the TNFα-induced phosphorylation of p38 and c-Jun N-terminal kinase (JNK) and impair the nuclear translocation of nuclear factor-κB (NF-κB) [18]. Inflammation plays a pivotal role in the pathomechanism of atherosclerosis and contributes to all of its stages, from plaque initiation to maturation and rupture [19]. In human endothelial cells, PGRN inhibits the atherosclerotic process induced by lipopolysaccharide through the activation of the Akt–eNOS pathway and the attenuation of the NF-κB pathway, resulting in the decreased expression of sVCAM-1, sICAM-1 and monocyte chemoattracant protein-1 (MCP-1) [20]. Moreover, the deletion of PGRN exacerbated atherosclerosis in ApoE knockout mice through the promotion of inflammation, accumulation of excessive cholesterol in macrophages and altered activity and amount of HDL-associated protein demonstrates the anti-atherogenic effect of PGRN. We found significant positive correlations between TNFα, sVCAM-1 and PGRN in HeFH patients and in the whole study population, which may indicate that vascular inflammation associated with enhanced atherogenesis in HeFH patients induces the expression of PGRN, which in turn attenuates the inflammatory process mediated by TNFα and sICAM-1. This hypothesis is supported by a recent study reporting that elevated serum PGRN associated positively with sVCAM-1 in patients with COVID-19. The authors concluded that high concentrations of PGRN may block TNFα-mediated inflammation in COVID-19 patients [21]. We found significant positive correlations between TNFα, sICAM-1 and PGRN in patients with type 2 diabetes and peripheral neuropathy, which also underlines the impact of PGRN on the inflammatory processes [22].

HDL particles are the first defensive barrier against the burden of high LDL-C [23]. The Framingham Heart Study proved the epidemiological association between low HDL-C and increased cardiovascular risk in a large population cohort [24]. HDL-C levels in FH populations have been extensively studied, but limited data are available on changes in the function, distribution and concentration of HDL subfractions within this special patient population. Beyond LDL cholesterol, high HDL cholesterol is one of the main markers of longer CVD risk-free survival in HeFH [25]. In contrast, most previous studies could not find differences between the HDL-C concentrations of HeFH patients and non-FH subjects [25,26]. However, former studies have reported enhanced chronic low-grade inflammation and systemic oxidative stress, as well as the impaired antioxidant and anti-inflammatory effects of HDL in HeFH patients. Moreover, enhanced cholesteryl ester transfer protein activity in HeFH leads to the triglyceride enrichment of HDL particles that are catabolized by an ApoE receptor pathway, resulting in a smaller HDL particle size [27]. The elevated concentrations of small pre-βHDL particles and lower levels of large HDL2 particles are also reported by another study [28]. Our results were in line with these data from the literature. The HDL-C concentration was similar in HeFH patients and the controls, but the concentrations and ratios of large HDL subfractions were significantly lower and the concentrations and ratios of small HDL subfractions were significantly higher in HeFH patients compared to the control population.

HDL functions, including the activities of HDL-associated enzymes, are often impaired in HeFH patients. HDL-associated PON1 hydrolyzes reactive oxygen species, resulting in decreased endogenous oxidative stress and the prevention of atherogenesis [29]. Some previous studies have observed decreased PON1 arylesterase activity in FH patients [30,31]. In contrast, we could not find significant differences between the PON1 paraoxonase, salt stimulated paraoxonase or arylesterase activities of HeFH patients and the controls, although the higher oxLDL and increased MPO activity demonstrated the increased oxidative stress in our HeFH population, which was similar to some other previous observations [32,33].

Previously, a significant negative correlation between HDL-C and PGRN was described in patients with myocardial infarction [34]. In our HeFH patients and in the whole study population, PGRN correlated positively with triglyceride and there was a negative correlation between HDL-C and PGRN in the whole study group, but not in HeFH patients. However, there were significant negative correlations between the ratio of large HDL and PGRN and significant positive correlations between the ratio of small HDL and PGRN in all participants, as well as each group separately. These correlations between PGRN and HDL subfractions are novel data. Furthermore, based on the multiple regression analysis, PGRN was best predicted by the small HDL subfraction, which supports the putative role of PGRN in HDL metabolism. Indeed, PGRN in mouse and human plasma is in a complex of ~180–190 kDa and has been found to be partially co-eluted with HDL by size-exclusion chromatography but not associated with lipoproteins; therefore, there is no evidence that PGRN is a component of HDL [35]. It must be noted that HDL-C itself does not determine serum PGRN. The possible role of PGRN in altered HDL metabolism needs further detailed investigation. 

Previous studies using carotid ultrasounds have demonstrated atherosclerotic plaque in 90% of asymptomatic patients with FH but without known atherosclerotic diseases [36]. PGRN has been found to be expressed in human atherosclerotic lesions [8]. Moreover, PGRN levels are significantly higher in patients with myocardial infarction compared to control subjects [34]. Therefore, we expected a higher serum concentration of PGRN in HeFH patients compared to normocholesterolemic subjects, but we could not find any significant differences between the two groups. We also compared our HeFH patients with AMI in their history to those who had not previously had AMI. Although serum PGRN was higher in HeFH patients with previous AMI, the difference was not significant, mostly because of the marked inter-individual variability of PGRN in the non-AMI group. Moreover, PGRN may be higher during the acute or subacute phase of AMI compared to the chronic phase. Further studies on a larger AMI population with long-term follow-ups are needed to clarify the kinetic changes of serum PGRN in AMI.

There is a considerable residual risk of clinical events in a large proportion of patients with atherosclerotic diseases, possibly driven in part by inflammatory processes. Lipid-lowering therapy used in HeFH may also modulate the inflammatory effects of atherosclerosis, both indirectly by attenuating LDL cholesterol mediated inflammation as well as directly by modulating inflammatory signaling. These mechanisms have been widely studied in statins and have also emerged in trials on PCSK9 inhibitors by enhancing the effects of statins [37]. It has been demonstrated that oxLDL upregulates PCSK9 in macrophages as well as inflammatory markers, including IL-1α, IL-6 and TNFα. These markers are reduced after inhibiting PCSK9, with small interfering RNA through NFκB-light-chain-enhancer of activated B cells [38]. The FOURIER trial proved that the hsCRP reducing efficacy of the PCSK9 inhibitor, evolocumab, is associated with absolute risk reduction in patients with stable atherosclerotic cardiovascular disease [39]. Recently, it has been reported that the administration of monoclonal antibodies against PCSK9 not only improves lipid profiles, but also beneficially alters inflammatory status and pulse wave velocity (PWV) in HeFH patients [40], although a significant reduction in PWV after a six-month PCSK9 inhibitor treatment was observed in FH subjects with a low neutrophil to lymphocyte ratio, indicating the higher inflammatory burden associated with atherosclerosis [41]. Accordingly, we must take advantage of the anti-inflammatory properties of the drugs already in use for HeFH and further investigate their role in the regulation of athero-inflammatory processes.

Some limitations of this study must be acknowledged. The relatively small number of studied individuals, especially patients with a positive history of cardiovascular complications, definitely reduced the power of our study; however, the results underline the potential importance of studying PGRN in this special patient population. Data on flow-mediated dilatation and arterial stiffness would improve our knowledge of the effects of PGRN on atherogenesis characterized by endothelial dysfunction.

## 5. Conclusions

To date, this is the first clinical study on the serum concentration of PRGN and its correlations to lipid parameters and the oxidative and inflammatory markers of atherosclerosis in an untreated HeFH population. Based on the strong correlation between HDL subfractions, inflammatory markers and PGRN and the lack of correlation with oxidative markers, our results suggest that PGRN may be involved in the anti-atherogenic response in HeFH patients as well as playing a role in the modification of HDL composition and the amelioration of inflammation, rather than as part of the response to oxidative stress. Further studies are needed to clarify the role of PGRN in regulating the atherosclerotic process in familial hypercholesterolemia. Understanding the role of PGRN in athero-inflammation may alter our therapeutic strategy for patients with familial hypercholesterolemia and may contribute to maximizing cardiovascular risk reduction in this patient population.

## Figures and Tables

**Figure 1 biomedicines-10-00771-f001:**
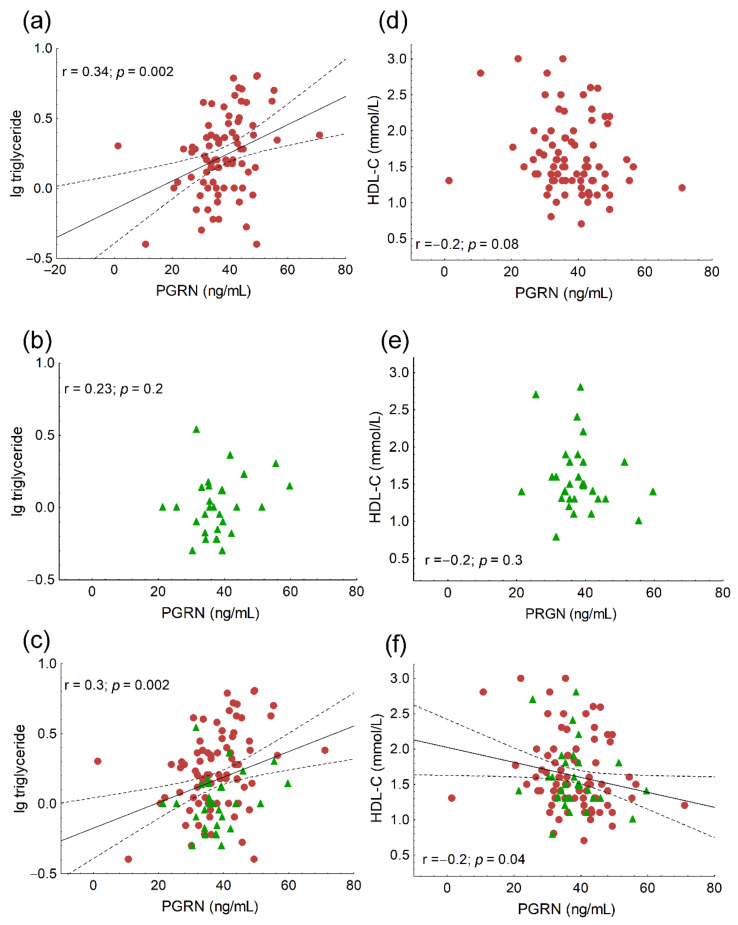
The correlations between serum triglyceride and progranulin (PGRN) in heterozygous familial hypercholesterolemic (HeFH) patients (**a**), healthy controls (**b**) and the whole study population (**c**). The correlations between HDL-C and progranulin (PGRN) in heterozygous familial hypercholesterolemic (HeFH) patients (**d**), healthy controls (**e**) and the whole study population (**f**).

**Figure 2 biomedicines-10-00771-f002:**
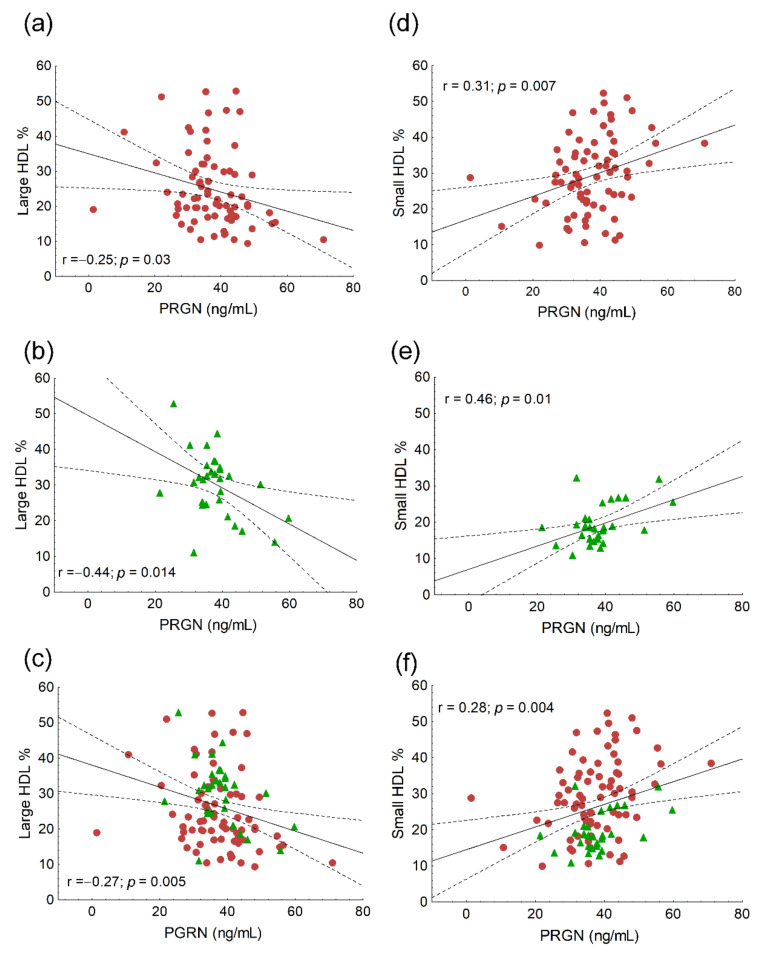
The correlations between the ratio of large HDL subfractions and serum progranulin (PGRN) in heterozygous familial hypercholesterolemic (HeFH) patients (**a**), healthy controls (**b**) and the whole study population (**c**). The correlations between the ratio of small HDL subfractions and serum progranulin (PGRN) in heterozygous familial hypercholesterolemic (HeFH) patients (**d**), healthy controls (**e**) and the whole study population (**f**).

**Figure 3 biomedicines-10-00771-f003:**
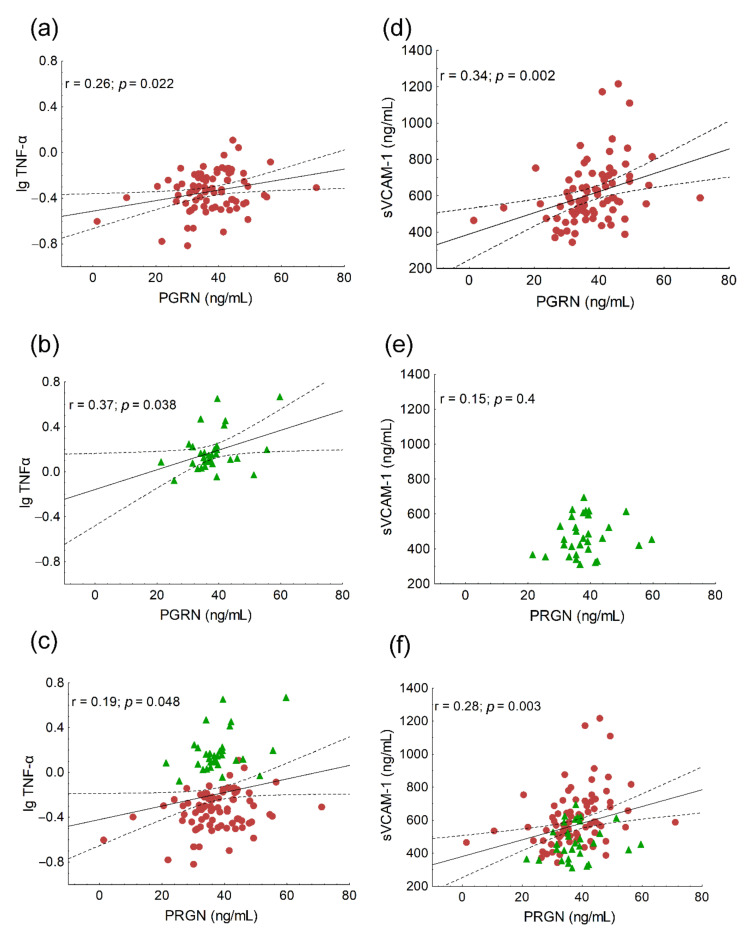
The correlations between the serum of tumor necrosis factor-alpha (TNFα) and progranulin (PGRN) in heterozygous familial hypercholesterolemic (HeFH) patients (**a**), healthy controls (**b**) and the whole study population (**c**). The correlations between the serum vascular intercellular adhesion molecule-1 (sVCAM-1) and progranulin (PGRN) in heterozygous familial hypercholesterolemic (HeFH) patients (**d**), healthy controls (**e**) and the whole study population (**f**).

**Figure 4 biomedicines-10-00771-f004:**
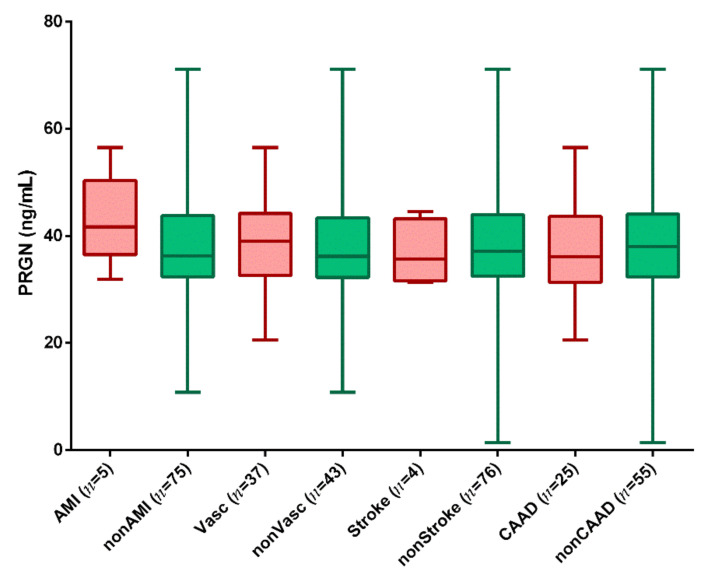
Boxplots showing serum progranulin (PGRN) in HeFH patients with and without previous acute myocardial infarction (AMI vs. nonAMI), with and without previous vascular complications (Vasc vs. nonVasc), with and without previous stroke (Stroke vs. nonStroke) and with and without carotid artery atherosclerotic disease (CAAD vs. nonCAAD). Box, mean ± SD; whiskers, mean ± 1.96 SD.

**Table 1 biomedicines-10-00771-t001:** The anthropometric and laboratory parameters of the study individuals. The values are presented as mean ± standard deviation or median (lower quartile–upper quartile).

	HeFH Patients	Controls	*p* Values
Number of subjects	81	32	
Male/Female	26/55	5/27	ns.
Age (years)	53.22 ± 14.5	41.8 ± 6.0	*p* < 0.001
**Lipid Parameters**			
Cholesterol (mmol/L)	8.87 ± 1.47	5.07 ± 0.78	*p* < 0.001
HDL-C (mmol/L)	1.62 ± 0.48	1.56 ± 0.46	ns.
LDL-C (mmol/L)	6.48 ± 1.28	2.93 ± 0.52	*p* < 0.001
Triglyceride (mmol/L)	1.6 (1.0–2.4)	1.0 (0.75–1.39)	*p* < 0.001
ApoB100 (g/L)	1.78 ± 0.38	0.94 ± 0.18	*p* < 0.001
ApoA1 (g/L)	1.71 ± 0.28	1.68 ± 0.31	ns.
Lp(a) (mg/L)	179 (75–857)	90 (30–214)	*p* < 0.05
**Inflammatory and Oxidative Markers**			
hsCRP (mg/L)	1.84 (0.70–2.90)	1.55 (0.6–2.95)	ns.
PON1 paraoxonase activity (U/L)	107.02 (43.61–166.5)	83.0 (47.9–167.4)	ns.
PON1 salt stimulated paraoxonase activity (U/L)	183.5 (103.2–322.6)	169.4 (97.3–297.4)	ns.
PON1 arylesterase activity (U/L)	143.2 ± 25.12	135.4 ± 36.8	*p* < 0.01
Myeloperoxidase (ng/mL)	297.7 (158.15–456.5)	135.7 (99.4–195.1)	*p* < 0.001
oxLDL (U/L)	187.98 ± 71.04	41.1 ± 9.57	*p* < 0.001
sICAM-1 (ng/mL)	270.66 ± 69.9	210.8 ± 32,2	*p* < 0.001
sVCAM-1 (ng/mL)	573.9 ± 140.45	467.7 ± 106.3	ns.
sCD40L (ng/mL)	10.02 ± 4.3	8.22 ± 3.44	ns.
ADMA (µmol/L)	0.66 ± 0.16	0.62 ± 0.17	ns.
TNFα (pg/mL)	0.47 ± 0.17	1.66 ± 0.91	*p* < 0.001
**Progranulin (ng/mL)**	37.66 ± 9.75	38.43 ± 7.74	ns.

Abbreviations: ApoA1, apolipoprotein A1; ApoB100, apolipoprotein B100; FH, familial hypercholesterolemia; HDL, high-density lipoprotein; hsCRP, high sensitive C-reactive protein; LDL, low-density lipoprotein; Lp(a), lipoprotein(a); PON1, paraoxonase-1; sCD40L, soluble CD40 ligand; sICAM-1, soluble intercellular adhesion molecule-1; sVCAM-1, soluble vascular adhesion molecule-1; TNFα, tumor necrosis factor alpha. ns.; not significant.

**Table 2 biomedicines-10-00771-t002:** The concentrations of lipoprotein subfractions in the study participants. The values are presented as mean ± standard deviation or median (lower quartile–upper quartile).

	HeFH Patients	Controls	*p* Values
**VLDL subfraction (%)**	19.76 ± 5.8	16.95 ± 2.2	0.01
**VLDL subfraction (mmol/L)**	1.77 ± 0.66	0.868 ± 0.17	<0.001
**Midband (IDL) (%)**	28.89 ± 4.5	29.83 ± 4.9	ns.
**Midband (IDL) (mmol/L)**	2.52 ± 0.62	1.505 ± 0.38	<0.001
**LDL subfractions**			
Large LDL (%)	27.3 ± 5.5	20.9 ± 5.8	<0.001
Small LDL (%)	3.2 (1.1–11.0)	0.5 (0–0.8)	<0.001
Large LDL (mmol/L)	2.29 (2.05–2.64)	1.047 (0.827–1.344)	<0.001
Small dense LDL (mmol/L)	0.18 (0.05–0.79)	0.026 (0–0.052)	<0.001
Mean LDL size (nm)	26.78 ± 0.58	27.26 ± 0.37	<0.05
**HDL subfractions**			
Large HDL (%)	24.7 ± 11.0	30.2 ± 8.9	0.02
Intermediate HDL (%)	46.0 ± 4.9	50.6 ± 4.7	<0.001
Small HDL (%)	29.3 ± 10.6	19.2 ± 5.4	<0.001
Large HDL (mmol/L)	0.35 (0.231–0.571)	0.453 (0.31–0.608)	<0.001
Intermediate HDL (mmol/L)	0.72 (0.613–0.932)	0.750 (0.659–0.853)	<0.05
Small HDL (mmol/L)	0.452 (0.374–0.523)	0.284 (0.246–0.336)	<0.01

Abbreviations: HDL, high-density lipoprotein; IDL, intermediate-density lipoprotein; LDL, low-density lipoprotein; VLDL, very low-density lipoprotein.

**Table 3 biomedicines-10-00771-t003:** The mean values of progranulin in FH patients with a low and high median of VLDL subfraction (%) and in FH patients with a low and high median of HDL-C compared to the controls. The values are presented as mean ± standard deviation.

	HeFH with Low VLDL (%) (<19.2%; *n* = 37)	HeFH with High VLDL (%) (≥19.2%; *n* = 37)	Controls (*n* = 32)	*p* Values (ANOVA)
Progranulin (ng/mL)	35.37 ± 8.2	39.66 ± 11.2	38.43 ± 7.74	0.13
	**HeFH with low HDL-C (mmol/l)** **(<1.5 mmol/l; *n* = 35)**	**HeFH with high HDL-C (mmol/l)** **(≥1.5 mmol/l; *n* = 44)**	**Controls (*n* = 32)**	
Progranulin (ng/mL)	39.32 ± 10.8	36.6 ± 9.1	38.43 ± 7.74	0.43

## Data Availability

The data presented in this study are available on request from the corresponding author.

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
