# Peer review of "Determination of Serum Progranulin in Patients with Untreated Familial Hypercholesterolemia"

_biomedicines, 2022, doi:10.3390/biomedicines10040771_

Round 1
Reviewer 1 Report
Determination of serum progranulin level in patients with newly diagnosed untreated familial hypercholesterolemia
I thank you for the opportunity to comment this study.
Authors conclusions were the following: Conclusions: Strong correlations between HDL subfractions, inflammatory markers and PGRN suggest that PGRN may exerts its anti-atherogenic effect in FH by alteration of HDL composition and by amelioration of inflammation rather than decreasing oxidative stress.
Comments
Main comment:
The association is not causal relationship. Authors needs re-word carefully the conclusions.
Other comments
I don’t see the purpose to mention that HeFH patients were newly diagnosed. Especially in the title this causes confusion.
Were the samples taken during acute visit to hospital or at some other timepoint. Please, clarify.
Add table 1. how many HeFH patients have had AMI at what age.
Results somehow doesn’t have the information regarding oxLDL. Please provide also figure regarding oxLDL that readers can see the lack of correlation.
Figures 1-3 are too small. Consider using colours.
The paragraph regarding HeFH and HDL subfractions needs to be extended.
Add limitations of this study.
The conclusion is that “A better understanding…” is at this timepoint overshooting. Kindly change.
Text is wild mixture of FH, HeFH and familial hypercholestrolemia – please make it consistent.
Author Response
Please find our response attached as a pdf file.

Reviewer 2 Report
In this interesting paper, the authors showed Strong correlations between HDL subfractions, inflammatory markers and PGRN suggest that PGRN may exerts its anti-atherogenic effect in FH by alteration of HDL composition and by amelioration of inflammation rather than decreasing oxidative stress. The article is interesting, well describe and the english style is adequate. However, I have several suggestions for the authors:
- In the Introduction Section, the authors should better explain the link between cholesterol and inflammation; please consider these manuscript (10.1161/JAHA.114.001403, 10.1007/s00592-019-01338-1) and comment these in the Introduction Section.
- In the Result Section, the authors showed a significant association between PGRN and TG; it may be interesting evaluate the correlation between PGRN and VLDL subfractions. If these data are not present, it may possible evaluate the correlation between VLDL and PGRN. If this correlation is significative, it may be interesting the analysis of PGRN value in FH patients divided in two groups according the median value of VLDL and controls (a novel table). As concerns the correlation of PGRN and HDL subfractions, it would also be interesting the analysis of PGRN in it may be interesting the analysis of PGRN value in FH patients divided in two groups according the median value of HDL and controls (a novel table). According these results, the discussion would be enriched by these findings.
- In the Discussion Section, please consider the possibility of novel studies to evaluate the role of PCSK9 inhibitors on this pathway (10.1016/j.atherosclerosis.2017.04.023). Previous studies showed interesting results on inflammatory pathway; please comment these studies (
10.1161/CIRCULATIONAHA.118.034032;
10.1007/s00592-021-01703-z;
10. 1194/ jlr. M0513 26;
10.1016/j.numecd.2021.08.034) and comment these in the Discussion Section.
Author Response

(The authors gave the same response as above.)

Round 2
Reviewer 2 Report
The authors adequately satisfied the requested revision